# Supplementation of Molasses-Based Liquid Feed for Cattle Fed on Limpograss Hay

**DOI:** 10.3390/ani12172227

**Published:** 2022-08-29

**Authors:** Daciele Abreu, José C. B. Dubeux, Luana Dantas Queiroz, David Jaramillo, Erick Rodrigo Da Silva Santos, Flávia van Cleef, Carlos Vela-Garcia, Nicolas DiLorenzo, Martin Ruiz-Moreno

**Affiliations:** 1Animal Science Department, Rural Federal University of Pernambuco, Recife 52171-900, PE, Brazil; 2North Florida Research and Education Center, University of Florida, Marianna, FL 32446, USA; 3U.S. Dairy Forage Research Center, USDA-ARS, 2615 Yellowstone Dr., Marshfield, WI 54449, USA; 4Department of Agricultural, Food and Nutritional Science, University of Alberta, 116 St. and 85 Ave., Edmonton, AB T6G 2R3, Canada

**Keywords:** digestibility, hemarthria, liquid feed, ruminal fermentation

## Abstract

**Simple Summary:**

Seasonality in forage production may limit the quantity and quality of forage. Even when forage is available in the period of scarcity, protein might still be a limiting nutrient in animal performance. Feed supplementation is a powerful tool to adjust nitrogen (N) levels in the diet of ruminants during critical periods. Urea is commonly used as a source of non-protein nitrogen in molasses supplements. This dietary protein provides amino acids as well as nitrogen for microbial protein synthesis. Moreover, molasses has organoleptic characteristics, such as palatability, increasing dry matter intake, through microbial growth, especially for fiber-digesting bacteria. The objective of this work was to evaluate the effects of different levels of liquid supplementation based on molasses enriched with 32% crude protein (as fed; 45% on DM basis) on ingestion, digestibility, and rumen fermentation. The results of this study indicated that addition of a molasses-based liquid feed to a *Hemarthria altissima* hay-based diet can improve the nutrient supply to animals; notably, volatile fatty acids (VFA) and microbial N. Liquid supplementation had an additive effect on intake of animals fed *Hemarthria altissima* hay but did not increase hay intake.

**Abstract:**

Two experiments were performed to evaluate the effects of (1) different levels of liquid supplementation (LS) based on molasses enriched with 32% (as fed; 45% on DM basis) crude protein (CP) on intake of *Hemarthria altissima* hay (LH), digestibility, and rumen fermentation, and (2) different levels of LS based on molasses enriched with 32% CP in the in vitro gas production in LH diets. In Exp. 1, twelve heifers and 12 adult male castrated and cannulated cattle were used. Treatments were allocated in a randomized block design, in four treatments: (CTL) access ad libitum to the LH; (SUP2) ad libitum access to LH and supplementation with 0.9 kg d^−1^; (SUP4) ad libitum access to LH and supplementation with 1.8 kg d^−1^ and; (SUP6) ad libitum access to LH with 2.7 kg d^−1^. In Exp. 2, treatments were carried out in a randomized block design with four different proportions of LH diet: (CTL) 100 LH, (SUP2) 85 LH and 15 LS, (SUP4) 70 LH and 30 LS, (SUP6) 55 LH and 45% LS. In Exp. 1, liquid molasses-based supplementation did not affect LH intake (*p* > 0.05). Molasses intake improved as the supply increased, not reducing the intake of LH. In Exp. 2, the addition of LS to the LH caused a change in the VFA profile, with an increase in propionate production in vitro.

## 1. Introduction

Worldwide, pastures and forages occupy 3.9 billion hectares, corresponding to approximately 30% of the non-polar land surface of the planet (FAO, 2006). Pastures are responsible for producing large amounts of digestible dry matter per area when treated and managed correctly. One of the focuses of pasture system is to promote the daily supply of forage capable of economically meeting the nutritional requirements of the herd. In order to achieve this goal, it is necessary to adjust the forage supply curve with the herd needs. Seasonality in forage production limits the capacity to adjust the supply curve, increasing the need for supplemental feeding to maintain the desired level of productivity [1].

Even when forage is available, in periods of scarcity, protein is often the most limiting nutrient in animal performance. In this context, supplementation acts as a tool to adjust deficient nitrogen levels in the diets of animals. Potential outcomes of protein supplementation include the increase in efficiency of degradation of the fibrous fraction and, consequently, the passage rate and dry matter intake of forage [2]. It is worth remembering that the basic condition for supplementation to work is the need for forage mass, so there is no limitation on intake, since the main objective of supplementation is to maximize the use of available forage [3]. Thus, supplementation with energy and protein sources for the cattle is crucial during critical periods of the year [4].

Several by-products of the industry of great nutritional value are available for feeding the cattle, providing an excellent opportunity to correct nutritional imbalances through the use of strategic supplements [5]. Sugarcane molasses, a rich source of sucrose, seems to be a viable option as a source of energy and minerals [6]. Molasses-based liquid supplementation has been used since the 1960s not only as a source of energy, but also as a source of sulfur and branched-chain volatile fatty acids, which are necessary nutrients for cellulolytic rumen microbes [7,8]. Molasses can supplement low quality forages by adding needed energy. Furthermore, molasses supplementation aids in microbial growth, especially for fiber-digesting bacteria, which can promote faster fiber degradation and lead to greater dry matter intake [9,10]. The acceptability of molasses can become a disadvantage as it can lead to over intake. Excessive intake of molasses might lead to negative effects on rumen fermentation, producing excess VFA and lactic acid, which in turn can decrease the pH, leading to acidosis [11]. However, this organoleptic characteristic, represented by its sweetening property and its flavor, helps in the ingestion of feed that do not always have good acceptability, such as urea [12].

Urea is commonly used as a source of non-protein nitrogen (NPN) in molasses supplements, mainly due to the reduced cost compared to plant-based protein supplements such as cottonseed meal and soybean meal [13]. Dietary protein plays an important role in ruminant nutrition, in addition to providing amino acids; it is also a source of nitrogen for the synthesis of microbial proteins [14]. Ammonia (NH_3_-N) can serve as a nitrogen source for microbial growth when carbohydrate is available as an energy source [15]. The optimal concentration of NH_3_-N for the maximum fermentation rate of forage-based diets is related to the availability of an energy-producing substrate [16]. Silva et al. [17], evaluating the effects of urea, molasses and their combinations, found that the urea + molasses combination resulted in a greater concentration of VFA. This result indicated that the concentration of short-chain volatile fatty acids (SCVFA) was greater when molasses was fed together with urea, compared to when supplied separately. It is believed that the greater concentration of SCVFA in this diet is due to the more complementary use of energy and N by the microorganisms, reaffirming the hypothesis that the timing of carbohydrate and N supplementation can improve fermentation. However, the presence of a supplement should stimulate forage intake and avoid a possible substitutive effect, aiming to maximize the ingestion rate, the digestion of the fibrous fraction, and the synthesis of microbial protein through the use of ammonia produced in the rumen [18,19].

The hypothesis of this study was that a liquid diet would have an additive effect on the intake and digestibility of limpograss hay, with an additive effect of molasses up to intermediate levels of supplementation. The objective of this work was to evaluate the effects of different levels of molasses-based liquid feed on intake, digestibility, and rumen fermentation.

## 2. Materials and Methods

### 2.1. Experiment 1

#### 2.1.1. Experimental Design, Animals, and Treatments

The study was conducted in June 2020 at the University of Florida Feed Efficiency Facility (FEF) located at the North Florida Research and Education Center in Marianna (30°46′35″ N, 85°14′17″ W). A total of 12 Angus heifers (400 ± 32 kg initial BW) and 12 adult male castrated and cannulated cattle (680 ± 115 kg initial BW), all of the Angus crossbreed, were used. On day 0, the animals were fasted for 16 h, being weighed and distributed in their respective pens. This distribution was performed by means of weight and sex in a randomized block design, over a period of 28 days in 4 treatments: (1) control (CTL) ad libitum access to *Hermathria altissima* hay (Limpograss hay) cv. Floralta; (2) SUP2, ad libitum access to limpograss hay (LH) cv. Floralta and supplementation with 0.9 kg d^−1^ molasses with 32% CP as fed; (3) SUP4, ad libitum access to LH cv. Floralta and supplementation with 1.8 kg d^−1^ molasses with 32% CP as fed; and (4) SUP6, ad libitum access to LH cv. Floralta with 2.7 kg d^−1^ molasses with 32% CP as fed. All the hay used in the experiment was purchased from limpograss growers from Florida matching the matured of stockpiled limpograss. The liquid supplement consisted of urea-fortified molasses with 32% CP on an as-fed basis (45% on a DM basis) and was provided by Quality Liquid Feed Inc. (Dodgeville, WI, USA). Animals were housed in individual pens at the FEF and had ad libitum access to water and LH, which was chopped using a Tub Grinder (Haybuster, Jamestown, ND, USA) and fed daily in the feed bunk. Diets were sent to a commercial laboratory (Dairy One Forage Laboratory, Ithaca, NY, USA) for nutrient composition analyses, which are presented in Table 1 (LH and liquid supplement). The amount of molasses-based liquid feed was weighed and offered daily to each animal individually in a plastic container inside the pen, separately from hay at a fixed time (745 h). Hay intake was measured using the GrowSafe^®^ system. Intake of liquid feed was recorded by weighing the amount offered and orts. Hay and liquid diet intake were recorded over a 4-day measurement period.

#### 2.1.2. Apparent Total Tract Digestibility

After 18 days of adaptation to diets and facilities, feed (hay and liquid supplement) and fecal samples were collected for 4 consecutive days in two periods of the day (8 h and 16 h), in order to determine the apparent total tract digestibility of dry matter (DM), organic matter (OM), CP, neutral detergent fiber (NDF), and acid detergent fiber (ADF). Feed samples were collected directly from the trough of each animal at three different points. Fecal samples were collected directly from the rectal ampoule. After collection, fecal and liquid feed samples were stored frozen at −20 °C. At the end of the experiment, hay and fecal samples were thawed and dried at 55 °C until constant weight was achieved, ground in a Wiley mill (Arthur H. Thomas Co., Philadelphia, PA, USA) to pass a 2-mm screen, and pooled within animals for further determination of nutrient content and digestibility marker concentration. Indigestible NDF (iNDF) was used as an internal indigestible marker [5,20].

#### 2.1.3. Ruminal Fluid and pH

Ruminal fluid was collected on day 32 every 3 h post-feeding for 24 h. Ruminal fluid was strained from a representative sample of digesta through 4 layers of cheesecloth, and pH was immediately measured using a manual pH meter (Corning Pinnacle M530, Corning Inc., Corning, NY, USA). Two samples of 10 mL were taken from each animal, and 1 mL of 20% (vol/vol) H2SO4 solution was added to stop fermentation. Ruminal samples were stored frozen at −20 °C for further analysis.

#### 2.1.4. Laboratory Analyses

For determination of sample DM and OM, approximately 0.5 g of hay and feces were weighed in duplicates, dried in a forced-air oven at 100 °C for 16 h, and subsequently ashed at 600 °C for 4 h. Approximately 0.5 g of hay and feces were weighed in duplicate into F57 bags (Ankom Technology Corp., Macedon, NY, USA), and incubated in the rumen of two cannulated cattle for 288 h [5,20]. The residue was analyzed for NDF concentration, using heat-stable α-amylase and sodium sulfite, and subsequent ADF as described by Van Soest et al. [21] in Ankom 200 Fiber Analyser (Ankon Tecnology Corp., Macedon, NY, USA) For CP concentration in feces and feed, samples were analyzed for total N using a CHNS analyzer by the Dumas dry combustion method (Vario Micro Cube; Elementar, Hanau, GER) following the official method [22].

The phenol-hypochlorite reaction was used to determine NH_3_-N concentrations as described by Broderick and Kang [23]. Post-incubation inoculum samples were centrifuged at 10,000× *g* for 15 min at 4 °C (Avanti J-E, Beckman Coulter Inc., Palo Alto, CA, USA). Absorbance was read at 665 nm in flat-bottom 96-well plates using a plate reader (DU 500; Beckman Coulter Inc.).

VFA concentrations in ruminal fluid were determined in a water-based solution using ethyl acetate extraction. Samples were centrifuged at 10,000× *g* for 15 min at 4 °C. Two milliliters of the supernatant were mixed with 0.4 mL (ratio 5:1) of a metaphosphoric acid:crotonic acid (internal standard) and the samples were frozen overnight. The next day, the samples were thawed and centrifuged again at 10,000× *g* for 15 min at 4 °C. The supernatant was transferred to 12 mm × 75 mm borosilicate disposable culture tubes (Fisherbrand; Thermo Fisher Scientific Inc., Waltham, MA, USA) and mixed with ethyl acetate to form a 2:1 ethyl acetate:supernatant mixture. The culture tubes were shaken vigorously and followed by a 5 min rest time to allow separation of the ethyl acetate. A subsample of ethyl acetate from the top layer was transferred to small vials prior to analysis. Samples were analyzed by gas chromatograph (Agilent 7820A GC, Agilent Technologies, Palo Alto, CA, USA) using a flame ionization detector and a capillary column (CP-WAX 58 FFAP 25 m × 0.53 mm, Varian CP7767; Varian Analytical Instruments, Walnut Creek, CA, USA). Column temperature was maintained at 110 °C, and the injector and detector temperatures were 200 °C and 220 °C, respectively. Samples were analyzed for total N using a CHNS analyzer via the Dumas dry combustion method (Vario Micro Cube; Elementary, Hanau, GER) coupled to an isotope ratio mass spectrometer (IsoPrime 100, IsoPrime, Manchester, UK) for δ15N analysis.

#### 2.1.5. Calculations and Statistical Analysis

Apparent total tract digestibilities of DM, OM, CP, NDF, and ADF were calculated as follows:100 − 100 × [(marker concentration in feed/marker concentration in feces) × (nutrient concentration in feces/nutrient concentration in feed)].

To estimate the proportion of N from limpograss and N from liquid diet in feces, the total N intake from each source was estimated by multiplying the DM intake from each source by the respective N content. Consequently, it was possible to calculate the N content of the total diet, as well as the proportional contribution of each source. The δ15N of the diet was calculated according to the following equation:δ^15^N_diet_ = [(δ^15^N_limpograss_ − δ^15^N_molasses_) × limpograss proportion N in diet] + δ^15^N_molasses_

To estimate the proportion of N from limpograss contained in feces, the calculation used was:$$\%\;limpograss\;N\;in\;feces=\frac{\left[\left(\delta^{15}Nfeces-∆feces-diet\right)-\delta^{15}Nmolasses\right]\times 100}{\left(\delta^{15}Nlimpograss-\delta^{15}Nmolasses\right)}$$

#### 2.1.6. Statistical Analysis

Data were analyzed as a generalized randomized block design. Means were grouped for each animal and analyzed using SAS mixed procedure. Treatments and their interactions were considered fixed effects. Animals in the treatment and sex were considered random effects. Contrasts were performed using orthogonal polynomials to verify linear and quadratic effects, as well as orthogonal contrast with supplement vs. without supplement. Means were compared using the SAS PDIFF adjusted for Tukey (*p* < 0.05). 

### 2.2. Experiment 2

#### 2.2.1. Treatments and Substrates

The evaluation of the in vitro fermentation parameters was conducted in a randomized block design using four different proportions of LH collected from Exp. 1 as a basal substrate plus increasing doses of molasses-based liquid feed (100:0, 85:15, 70:30, or 55:45). Treatments consisted of 1) CTL, 0.7707 g hay (100:0); 2) SUP2, 0.6551g hay + 0.1567 g of liquid supplementation (85:15); 3) SUP4, 0.5395 g hay + 0.3134 g of liquid supplementation (70:30); and 4) SUP 6, 0.4239 g hay + 0.4701 g of liquid supplementation (55:45). Treatments were chosen to mimic the LH and molasses enriched with urea proportion used in a previous in vivo study and total organic matter was similar across treatments. Substrates were sent to a commercial laboratory (Dairy One Forage Laboratory, Ithaca, NY) for nutrient composition analyses, which are presented in Table 1 (LH and liquid supplement).

#### 2.2.2. In Vitro Incubations

Ruminal fluid was collected from two ruminally cannulated crossbred steers (average BW of 630 kg). The animals consumed Tifton 85 bermudagrass hay ad libitum for at least 2 weeks before the ruminal fluid collection. A 500 mL bottle per animal was collected, which consisted of filtering representative samples of the digesta in four layers of cheesecloth and transported to the laboratory within 30 min of collection. Ruminal fluid collected from each animal was mixed in equal proportions. A 3:1 ratio of saliva and rumen fluid was used for all incubations [24]. Each bottle contained 0.7 g of dry matter and received 50 mL of ruminal inoculum. Substrates and treatments were incubated for 48 h at a temperature of 39 °C and constant agitation (60 rpm) in 125 mL glass bottles. Three incubations were carried out, starting on three different and consecutive days. Each day of incubation, four bottles per treatment, two blanks (only ruminal fluid and saliva), and two bottles with the standard sample (*Cynodon dactylon*) were used, totaling 20 bottles per incubation. Initial pH was recorded at the beginning of each incubation period. Final pH was recorded, and fermentation was halted by adding 0.5 mL of a 20% H2SO4 solution to each bottle. Three 3 mL samples were collected from each bottle and stored at −20 °C for subsequent VFA and NH_3_N analysis. Concentrations of NH_3_-n and VFA in the incubation fluid were measured as described in Exp. 1. The final pressure of each bottle was measured using a digital manometer.

#### 2.2.3. In Vitro Organic Matter Digestibility (IVOMD)

Dry matter concentration was determined by weighing 1 g of ground sample and drying at 105 °C for 16 h, and OM was determined by ashing samples at 600 °C for 6 h. The IVOMD was performed as described by Tilley and Terry [25], with the adaptations described by Holden [26], and 0.7 g of DM substrate was incubated with 50 mL of McDougall saliva in a 3:1 ratio (saliva:ruminal fluid) in 100 mL plastic centrifuge tubes for 48 h under constant agitation (60 rpm). Two tubes per treatment, one blank tube (no substrate or treatment) and one standard tube (*Cynodon dactylon*), were incubated on each of the 3 days of incubation. After the initial 48 h, 6 mL of HCl were added to the tubes along with 2 mL of a 5% pepsin solution. Tubes were then incubated for an additional 48 h. The samples were then filtered through P8 filters (Fisherbrand; Thermo Fisher Scientific Inc.). Filters with wet samples were then dried at 105 °C in a forced air oven for 24 h to determine the DIVMS. The dried samples were then placed in a muffle for 6 h at 650 °C. The ash was then placed in an oven at 105 °C for 24 h before recording the weight.

#### 2.2.4. Statistical Analysis

Total final pressure, IVOMD, NH_3_-N, final pH, and VFA data were analyzed as a randomized complete block design using the SAS MIXED procedure. Treatments were considered fixed effect and day (block) random effect. An average of 4 bottles per day was considered the experimental unit with 3 replications (days). Orthogonal polynomial contrasts were performed using the SAS MIXED procedure to determine the linear, quadratic, and cubic effects of the liquid diet inclusion on fermentation parameters. Means were compared using the SAS PDIFF adjusted for Tukey (*p* < 0.05).

## 3. Results

### 3.1. Experiment 1

#### 3.1.1. Diet Intake

Molasses-based liquid supplementation did not affect limpograss hay intake (DM and OM) (*p* > 0.05) (Table 2). Molasses intake improved as the supply increased, not reducing the intake of LH. CP intake had a treatment effect (*p* < 0.0001), showing a positive linear response (*p* < 0.0001) to levels of supplementation with liquid diet. Neutral detergent fiber intake had positive linear growth with increasing doses of supplementation (*p* < 0.05). In the current study, NDF intake was greater for the maximum supplementation treatment, although hay intake did not differ between treatments.

#### 3.1.2. Nitrogen Uptake

Nitrogen proportion in LH and liquid supplementation found in feces (estimated by δ^15^N) are shown in Table 3. There was a treatment effect (*p* < 0.0001), where it was possible to identify a greater presence of nitrogen from the liquid supplementation in treatments SUP4 and SUP6, with a quadratic effect (*p* < 0.0001).

#### 3.1.3. Digestibility of Diets

There was no statistical difference (*p* > 0.05) for DM, OM, and NDF digestibility (Table 4). For CP, there was a linear (*p* < 0.0001) and quadratic (*p* = 0.0003) effect, where in the treatment CTL, without the presence of molasses, the digestibility was 34 percentage points lower when compared to the other treatments. Although NDF digestibility was not affected by supplementation levels, the same was not true for ADF (*p* < 0.05). The control treatments and those with the highest supplementation level (SUP6) had a decrease in relation to the others.

#### 3.1.4. Dynamics of In Vivo Ruminal Fermentation Indicators


*pH*


Ruminal pH was not affected by different concentrations of liquid diet (*p* = 0.109; Figure 1). There was an interaction between treatment × time (*p* = 0.013), but there were no differences between means. The lowest pH values occurred at 23 h (6.62) and the highest peak at 11 h (6.84). Although ruminal pH had a quadratic effect (*p* = 0.029), values were never lower than 6.0, even at the highest level of supplementation, indicating that cellulolytic bacteria activity was not negatively affected [27].


*Ammonia (NH_3_-N)*


As expected, there was a treatment effect for NH_3_-N (*p* = 0.003), where the treatments with protein supplementation had a higher concentration of ammonia when compared to control treatment (Figure 2). Mean ammonia concentration of supplemented treatments was 4.6 mM. All treatments with liquid diet had similar NH_3_-N concentrations until the supplementation offer time (740 h). The highest concentration of NH_3_-N occurred during the 5 h after feeding, reflecting the amount of urea ingested with the molasses in the diets soon after feeding.


*Volatile fatty acids*


Molar proportions of acetic, propionic, isobutyric, butyric, isovaleric, valeric, and total VFA according to the treatments adopted are shown in Table 5. No differences were observed (*p* > 0.05) in molar concentrations of acetic acid, between treatments. For propionic acid, no differences were observed between treatments (*p* = 0.219). However, an inter-action between time × treatment (*p* < 0.05) was observed, but there were no differences be-tween means. Increasing concentrations of liquid diet had no effect for butyrate or valerate (*p* > 0.005). For isobutyrate, time effect (*p* = 0.004) and quadratic effect (*p* = 0.033) were ob-tained with no difference between the means. However, there was an interaction between time × treatment (*p* = 0.021), where, at 8 h, the SUP6 treatment stood out over the others, with molar ratio values of 2.6 mol/100 mol (Figure 3A). Molar proportions of isovalerate had a treatment effect (*p* = 0.005, Figure 3B). The supplementation treatment with 1.8 kg d^−1^ of molasses with 32% CP (SUP4) stood out in relation to the others, followed by the SUP2 and SUP6 treatment. There was also time × treatment interaction (*p* = 0.004). The control treatment obtained, at 5:00 am in the morning, a lower molar proportion of isovaleric acid (0.7 mol/100 mol) in relation to the others, and the SUP4 treatment obtained a higher proportion with 1.4 mol/100 mol. Molar proportions of VFAs were not affected by treatments.

### 3.2. Experiment 2

#### Dynamics of In Vitro Ruminal Fermentation Indicators


*Gas production*


The effects of increasing doses of liquid supplementation on the in vitro fermentation parameters and total gas production are presented in Figure 4. There was a quadratic effect (*p* < 0.05) for the total gas production.

Control treatment had greater gas production per unit of digestible organic matter (Figure 5). Treatments with liquid diet did not differ from each other and were smaller than the control treatment.


*In vitro rumen fluid pH*


Ruminal pH is represented in Figure 6. There was a treatment effect (*p* = 0.0008) and a linear increasing effect (*p* = 0.0001) as liquid diet concentrations were increased.


*Ammonia (NH_3_-N)*


NH_3_-N concentrations are observed in Figure 7. There was a treatment effect and a linear increasing effect (*p* < 0.05). This study showed that the ruminal concentration of NH_3_-N increased along molasses-based liquid supplement levels (32% CP), ranging from 4.8 to 20.2 mM.


*Volatile fatty acids*


There was a decreasing linear effect for acetate (*p* = 0.011), isobutyrate (*p* = 0.0002) and isovalerate (Table 6). There was a treatment effect (*p* = 0.013) and an increasing linear effect (*p* = 0.002) for the molar proportions of propionate. There was no significant difference in butyrate levels in the present work (*p* = 0.108). Total concentrations between treatments with urea ranged from 90.7 to 105.3 mM. 

## 4. Discussion

### 4.1. Experiment 1

#### 4.1.1. Diet Intake

Limpograss intake was not affected by molasses-based liquid supplementation. These results are in agreement with [28] who, evaluating the interaction effects between forage and different types of supplementation, did not obtain supplementation effects for DM and OM intake of forage. According to [1], depending on the amount of energy supplement used, animal intake may be reduced because the soluble sugars present reduce ruminal pH and consequently reduce fiber digestibility. Molasses intake improved as the supply increased, not reducing the intake of LH. Forage intake was constant at different levels of liquid supplementation, which indicates, at different levels of supplementation, that there was an additive effect of total intake on the corresponding levels of molasses provided [3]. The use of low quality forages, with NDF concentration above 60% and CP below 6%, can reduce intake, as a result of the low digestibility of the fibrous fraction [17]. In the present experiment, the ruminal pH values (Figure 1) were always above 6.5 for all supplementation levels, not representing a limiting factor for forage intake, which may be a justification for the non-difference between intake [29]. CP intake had a treatment effect to levels of supplementation with liquid diet. The increase in feed intake with an increase in the protein content in the diet, when it occurs, is generally due to greater fermentative activity in the rumen, generating greater production of microbial protein. [30].

In the current study, NDF intake was greater for the maximum supplementation treatment, although hay intake did not differ between treatments. According to [31], NDF is the fibrous fraction of feed that is not digested or insoluble when in contact with a neutral detergent solution. NDF is considered an important factor that affects the quality of diet. This fraction is inversely related to intake and the available energy content of the feed, due to its slow rate of passage in the reticulum-rumen, which results in a reduction on intake of total DM [32]. However, chemical composition of NDF, as well as the activity of the ruminal microbiota, directly affect the parameters of the dynamics of degradation and transit through the digestive tract, which are responsible for the effect of NDF on nutrient intake and digestion [33]. In the present study, the supplementation with nitrogen compounds at a higher level allowed an increase in the CP content, which may have led to the maximization of the use of the forage fibrous fraction by the ruminal microbiota [34].

#### 4.1.2. Nitrogen Uptake

The proportion of N excreted in feces was strongly related to the supply of liquid supplement in the diet. In the present study, CP digestibility ranged from 60% for control diets and 95% for diets with 2.7 kg day^−1^ of liquid supplementation (Table 3). In ruminants, a fecal protein loss of up to 30% or more is common [35]. According to [36], about 69% of the N found in urine comes from urea. Fecal urea excretion was quantified by [37] and the combination of urea and ammonia represents approximately only 9% of the total N excreted [37]. This implies that fecal N in diets is mainly microbial in origin with lower amounts of undegraded dietary protein and endogenous secretions [38]. Additional endogenous N secretions and/or losses along the gastrointestinal tract are a result of epithelial cell desquamation and mucus secretion. It is important to remember that the inclusion of non-degradable fiber in the diet of ruminants results in more digesta passing through the small intestine; as a result, more epithelial cells are released [39]. For a better understanding of the utilization and excretion of urea offered in the diet, more studies on urea flow need to be carried out, following all excretion mechanisms (e.g., urine, feces, milk).

#### 4.1.3. Digestibility of Diets

Liquid feed supplementation can improve ruminal fiber digestibility, and the magnitude of this response is linked to forage quality; in other words, increased digestibility due to liquid supplementation is more easily noticeable with low quality forage [28]. CP digestibility without the presence of molasses was 34 percentage points lower when compared to the other treatments. According to [17], this may indicate that molasses energy was used to release protein in the diet content, resulting in greater true CP digestibility.

For ADF, the control treatments and those with the highest supplementation level had a decrease in relation to the others. One possible explanation is that added sugars to forage-based diets can decrease fiber digestion [40]. On the other hand, moderate levels of supplementation increased FDA digestibility. In the control treatment, the lower value for ADF digestibility may have occurred due to the lower amount of N available to the microbes that ferment the slower-digesting fiber. However, in SUP6, the increased availability of carbohydrate readily available in the presence of available N in the rumen may have had direct inhibitory effects on some ruminal microbes [41]. In general, discrepancies in the literature regarding the response in fiber digestibility to molasses supplementation can be explained by the wide variability in molasses sources, supplementation levels, and forage quality [6].

#### 4.1.4. Dynamics of In Vivo Ruminal Fermentation Indicators


*In vitro rumen fluid pH*


Ruminal pH values were never lower than 6.0, even at the highest level of supplementation, indicating that cellulolytic bacteria activity was not negatively affected [27]. Ref. [42] evaluated the effect of replacing wheat bran by graded levels of urea-enriched molasses in ruminal fermentation, and also found no significant difference in pH levels. Soluble carbohydrates are fermented quickly in the rumen, lowering ruminal pH. However, this variation in supplementation levels in the study may have been low enough that large variations in ruminal pH were not observed, which would support the lack of differences in nutrient digestibility [6].


*Ammonia*


It is important to point out that ammonia concentration in the rumen fluid can vary from 1 to 40 mM, but values lower than 3.57 mM can compromise microbial growth [29,43]. Mean ammonia concentration of supplemented treatments was 4.6 mM. All treatments with liquid diet had similar NH_3_-N concentrations until the supplementation offers time (740 h). The highest concentration of NH_3_-N occurred during the initial 5 h after feeding, reflecting the amount of urea ingested with the molasses in the diets soon after feeding. A similar result was observed by [44], who, evaluating the effects of bismuth subsalicylate and encapsulated calcium-ammonium nitrate, alone and in combination, obtained the same behavior, where all treatments had similar peak and concentrations of NH_3_-N for 4 h straight after feeding, in relation to the treatment without a non-protein nitrogen source [45]. Testing different proportions of urea-enriched molasses showed that the highest concentration of NH_3_-N occurred 4 h after feeding, reflecting the amount of urea ingested with molasses in diets right after the presentation. According to [46], a balance between intake of hay and supplement diets by ruminal microorganisms may have resulted in an increase in the ruminal ammonia concentration after feeding.


*Volatile fatty acids (VFA)*


Ref. [47] reported that different protein levels did not affect ruminal pH, total VFA, acetate, propionate, and butyrate concentrations. This behavior was the opposite found by Ref. [46], who obtained a linear decrease in acetate as the proportions of the mixture of molasses and glycerol increased. For products generated from ruminal fermentation, acetate is the most oxidized and its presence results in maximum energy yield for the bacteria [29]. Ref. [48] states that increasing the digestibility of a feed increases the proportion of propionate in the final fermentation products formed from it. There was no difference in digestibility between treatments (Table 4), which may explain the similar values of propionic acid. Unless grain makes up the majority of the diet, forage and grain mixtures result in propionate production similar to forage-only diets [48].

Different responses were found, where the increase in urea in the diet resulted in a decrease in the proportion of these acids. However, values found by these authors (1.77 and 2.15 mol/100 mol), in general, were greater than the average values found in the present study (0.55 and 0.9 mol/100 mol) for isobutyrate and isovalerate, respectively [49]. Ref. [17] reported molar concentrations of 0.03 mol/100 mol for isobutyrate and 0.05 mol/100 mol for isovalerate. Isobutyric and isovaleric acids belong to the group of branched-chain volatile fatty acids and are indicative of the ruminal fermentation of amino acids, when, in high concentrations, they favor the accumulation of VFAs, resulting in a reduction in pH, which justifies the maintenance of ideal values for pH in different treatments in this study (Figure 2) [50].

Molar proportions of VFAs were not affected by treatments. A similar answer was found by [6], who, evaluating the effects of pasture supplementation with molasses and corn derivatives, had no differences in the molar proportions of the different treatments. In contrast, Ref. [49] obtained significant linear growth with increasing levels of urea in the diet. Final DM intake between diets was not significant. It is possible that the carbohydrate composition of the diets was similar and urea levels were not high enough to improve microbial growth conditions in the rumen. Results in similar molar ratios for all levels of liquid supplementation offered

### 4.2. Experiment 2

#### Dynamics of In Vitro Ruminal Fermentation Indicators


*Gas production*


The quadratic effect in total gas production indicates the importance of including the liquid supplement on the fermentability of the diet [46]. A similar result was obtained evaluating the effects of increasing doses of liquid molasses and raw glycerol supplementation on the parameters of rumen fermentation, obtaining a quadratic effect in increasing the proportion of the inclusion of the mixture. This quadratic effect indicates that, from a certain point onwards, there is a decrease in gas production. Propionate fermentation, in a higher concentration, in SUP6 treatment (Table 6) may have caused the tendency to decrease the total gas production [51]. Ref. [52] found that gas is mainly produced when the substrate is fermented into acetate and butyrate. When gas is fermented into propionate, less gas is produced. In the present study, acetate levels did not differ between treatments and butyrate levels differed only for the diet with the highest supplement concentration, which may justify the similar gas volume between treatments. A reduction in gas production due to added glycerol was also observed by [51], possibly because the end product of glycerol fermentation was mainly propionate, which produces less gas than acetate, presenting that molasses fermentation had high gas production when associated with sucrose concentration. Contrary results were found in the present study, where molasses associated with a protein source resulted in a decrease in the volume of gases per unit of digestible organic matter.


*Ruminal pH*


The linear increasing effect as liquid diet concentrations was increased was the opposite of those found by Ref. [17], who state that diets containing molasses tend to lower the pH. However, this increase in ruminal pH may be associated with the presence of urea in the liquid supplement, which is rapidly hydrolyzed by bacterial urease, resulting in ammonia production (Figure 6), raising the pH [53]. Ref. [54] evaluated four levels of urea in diets for dairy cows and observed an increase in pH after the period of diet offer. Urea is a source of nitrogen for rumen microbes; in addition, dietary urea can also have an alkalizing effect in the rumen. This means a decrease in ruminal pH drops in the presence of highly fermentable diets [54]. Similar results were obtained by Ref. [55], which evaluated the effect of increasing the concentration of degradable protein intake in the diet through the inclusion of urea in beef cattle, also obtaining a linear increasing effect on pH with increasing concentrations of urea in rumen. Finally, it is important to remember that the present study was carried out under in vitro conditions, where no animal absorption occurs, leading to accumulation of fermentative variables during fermentation [46].


*Ammonia*


It was reported that ruminal NH_3_-N concentration increased when protein supplementation was increased [56]. Ref. [55] also observed higher ammonia concentrations for animals fed urea sources compared to the control diet. According to [17], urea-containing diets had higher total N flux and NH_3_-N, and the study’s urea supplementation provided additional N compared to other diets without a protein source. Ruminal concentration of NH_3_-N is extremely important in ruminant animals, as it is a source of bacteria for the synthesis of its microbial protein, providing a source of high quality protein for the host animal [57].


*Volatile fatty acids*


Researchers reported lower isobutyrate concentrations when cattle had ad libitum access to a liquid molasses-urea supplement [58]. Increases in molar proportions of propionate and butyrate and decreases in molar proportions of acetate are in line with previous research in which increasing amounts of molasses promote the proliferation of VFA-producing microorganisms [59].

There are several studies that confirm that the ruminal fermentation of molasses increases butyrate [60,61]; however, there was no significant difference in butyrate levels in the present work. Ref. [51] also found no differences in butyrate concentrations in molasses-based diets. For total concentrations of VFA, there was a treatment effect and increasing linear effect, where the treatments of 30% and 45% enrichment had higher concentrations compared to the others. Total concentrations between treatments had urea ranging from 90.7 to 105.3 mM. The total concentrations of VFA found by [17] averaged 69.2 mM. These differences in total VFA concentrations can be explained by the fact that this experiment was carried out under in vitro conditions, where no absorption occurs, leading to the accumulation of VFA produced during fermentation [46]. Ref. [17] tested the use of molasses with and without urea, which obtained similar results to the present work, where the total concentration of VFA was significantly better for the molasses associated with urea. This fact supports the hypothesis that increasing carbohydrate and nitrogen sources in the diet can improve fermentation.

## 5. Conclusions

It was hypothesized that the liquid diet would have an additive effect on LH intake and digestibility, with this additive effect occurring up to intermediate levels of supplementation. In contrast to our hypothesis, however, the levels tested in this study did not improve the use of *Hermathria altissima* as assessed in digestibility. The results of this study indicated that the addition of urea and molasses to a limpograss hay-based diet can improve the nutrient supply to animals, notably VFA supply and microbial N supply. Ruminal AGV and microbial N are indicators of rumen function, which may indicate a better rumen environment; however, they did not result in higher digestibility. Molasses-based liquid supplementation with 32% CP had an additive effect in animals fed LH hay. Perhaps a larger number of animals during future studies will help to achieve clearer results regarding different levels of supplementation. Future studies should focus on exploring different levels of liquid diet that help to increase total intake.

## Figures and Tables

**Figure 1 animals-12-02227-f001:**
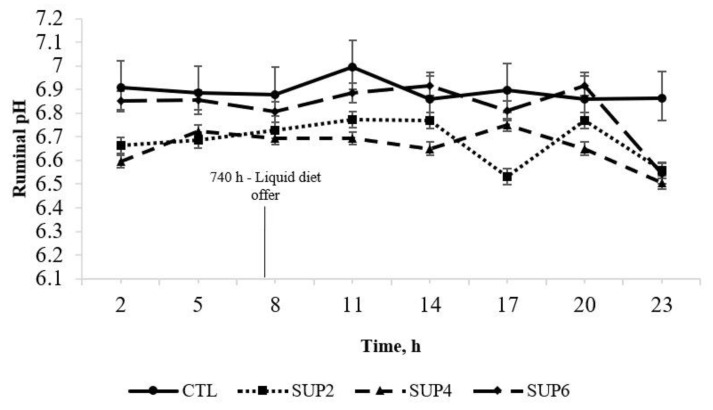
Effects of different levels of liquid molasses supplementation enriched with 32% CP on pH levels of LH-based diets ad libitum for 24 h, every three hours. A treatment × time interaction (*p* = 0.013) was observed. CTL = Ad libitum access to LH cv. Floralta. SUP2 = Ad libitum access to LH cv. Floralta and supplementation with 0.9 kg d^−1^ molasses with 32% CP. SUP4 = Ad libitum access to LH cv. Floralta and supplementation with 1.8 kg d^−1^ molasses with 32% CP. SUP6 = Ad libitum access to LH cv. Floralta and supplementation with 2.7 kg d^−1^ molasses with 32% CP. Error bars represent the SE of treatment differences for treatment × time interaction.

**Figure 2 animals-12-02227-f002:**
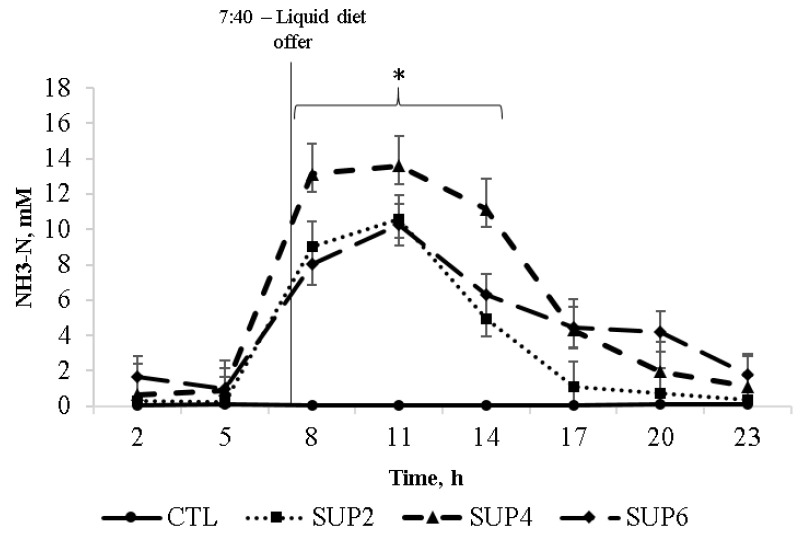
Effects of different levels of liquid supplementation based on molasses enriched with 32% CP on NH_3_-N concentrations in diets based on LH ad libitum for 24 h, every three hours. There was treatment effect (*p* = 0.003) and time (*p* < 0.0001) and treatment × time interaction (*p* < 0.0001). CTL = Ad libitum access to LH cv. Floralta. SUP2 = Ad libitum access to LH cv. Floralta and supplementation with 0.9 kg d^−1^ molasses with 32% CP. SUP4 = Ad libitum access to LH cv. Floralta and supplementation with 1.8 kg d^−1^ molasses with 32% CP. SUP6 = Ad libitum access to LH cv. Floralta and supplementation with 2.7 kg d^−1^ molasses with 32% CP. Error bars represent the SE of treatment differences for treatment × time interaction. * Means on times 8, 11, and 14 h were significantly different within time using the PDIFF procedure from SAS adjusted by Tukey (*p* < 0.05).

**Figure 3 animals-12-02227-f003:**
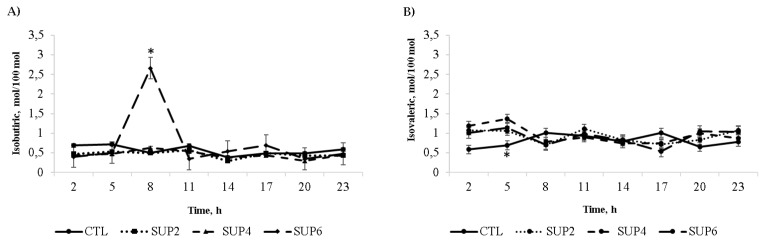
(**A**) Effects of different levels of liquid supplementation based on molasses enriched with 32% CP on ruminal concentrations of isobutyric acid in LH-based diets ad libitum for 24 h, every three hours. There was treatment effect (*p* = 0.004) and treatment × time interaction (*p* = 0.002), and quadratic effect (*p* = 0.033); (**B**) effects of different levels of liquid supplementation based on molasses enriched with 32% CP on ruminal concentrations of isovaleric acid in LH-based diets ad libitum for 24 h, every three hours. There was treatment effect (*p* = 0.031) and treatment × time interaction (*p* = 0.004), and quadratic effect (*p* = 0.027). CTL = Ad libitum access to LH cv. Floralta. SUP2 = Ad libitum access to LH cv. Floralta and supplementation with 0.9 kg d^−1^ molasses with 32% CP. SUP4 = Ad libitum access to LH cv. Floralta and supplementation with 1.8 kg d^−1^ molasses with 32% CP. SUP6 = Ad libitum access to LH cv. Floralta and supplementation with 2.7 kg d^−1^ molasses with 32% CP. Error bars represent the SE of treatment differences for treatment × time interaction. * Means were significantly different within time using the PDIFF procedure from SAS adjusted by Tukey (*p* < 0.05).

**Figure 4 animals-12-02227-f004:**
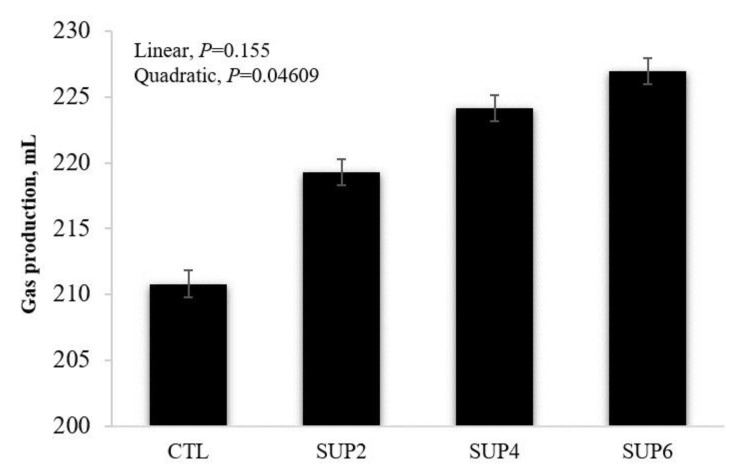
Effects of different levels of molasses-based liquid supplementation (32% CP) on gas production, evaluated 48 h after incubation in vitro in laboratory. CTL = 100% LH, SUP2 = 85% LH and 15% liquid supplementation, SUP4 = 70% LH and 30% liquid supplementation, SUP6 = 55% LH and 45% liquid supplementation. Error bars represent the SE for treatments.

**Figure 5 animals-12-02227-f005:**
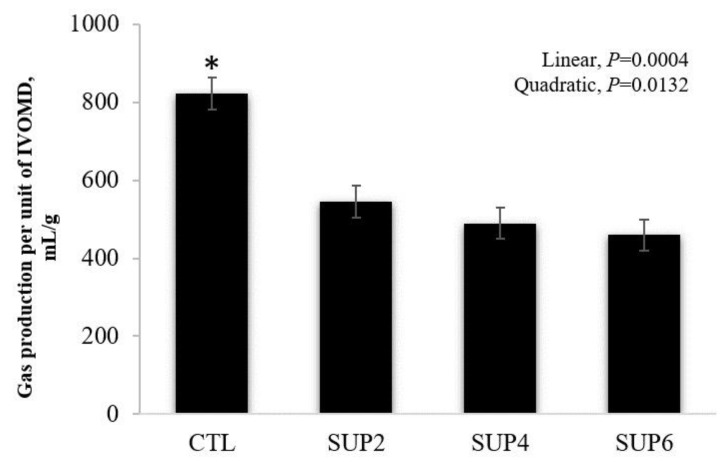
Effects of different levels of molasses-based liquid supplementation (32% CP) on gas production per unit of digestible organic matter, evaluated 48 h after incubation in vitro in the laboratory. CTL = 100% LH, SUP2 = 85% LH and 15% liquid supplementation, SUP4 = 70% LH and 30% liquid supplementation, SUP6 = 55% LH and 45% liquid supplementation. Error bars represent the SE for treatments. * Significantly different using the PDIFF procedure from SAS adjusted by Tukey (*p* < 0.05).

**Figure 6 animals-12-02227-f006:**
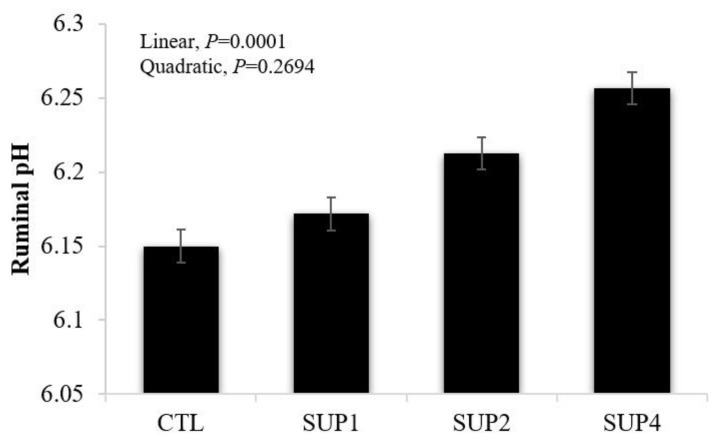
Effects of different levels of molasses-based liquid supplementation (32% CP) on pH levels, evaluated 48 h after incubation in vitro in the laboratory. CTL = 100% LH, SUP2 = 85% LH and 15% liquid supplementation, SUP4 = 70% LH and 30% liquid supplementation, SUP6 = 55% LH and 45% liquid supplementation. Error bars represent the SE for treatments.

**Figure 7 animals-12-02227-f007:**
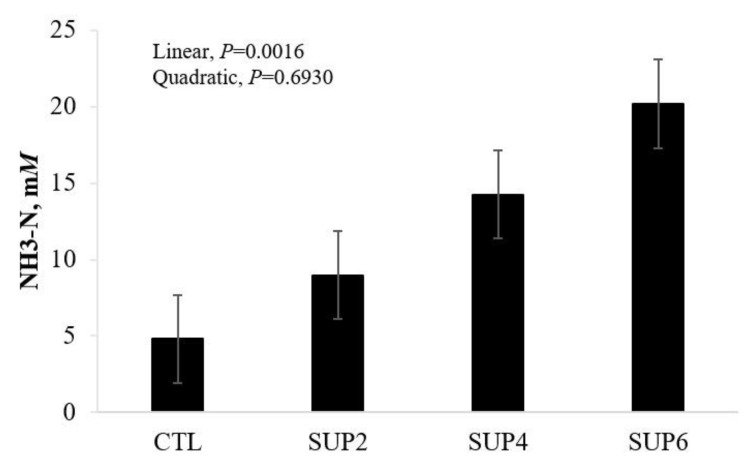
Effects of different levels of molasses-based liquid supplementation (32% CP) on NH_3_-N concentrations, evaluated 48 h after incubation in vitro in the laboratory. CTL = 100% LH, SUP2 = 85% LH and 15% liquid supplementation, SUP4 = 70% LH and 30% liquid supplementation, SUP6 = 55% LH and 45% liquid supplementation. Error bars represent the SE for treatments.

**Table 1 animals-12-02227-t001:** Analyzed chemical composition of limpograss hay and liquid supplement fed to beef heifers and steers ^1^.

Item	Hay	Liquid Supplement ^2^
DM%	92.9	68.8
OM, %DM	64.5	57.9
CP, %DM	3.9	45.5
NDF, %DM	79.1	-
ADF, %DM	42.6	-
TDN, %DM	54	73
Calcium, %DM	0.2	1.30
Phosphorus, %DM	0.17	1.56
Magnesium, %DM	0.13	0.47
Potassium, %DM	0.68	4.28
Sodium, %DM	0.042	0.136
Sulfur,%DM	0.14	1.61

^1^ Analyzed by a commercial laboratory using a wet chemistry package (Dairy One, Ithaca, NY, USA). ^2^ Liquid supplementation based on molasses enriched with 32% CP.

**Table 2 animals-12-02227-t002:** Effects of different levels of liquid supplementation based on molasses-based liquid feed (32% CP) on the intake of diets with limpograss hay ad libitum.

	Treatments ^1^					Contrast *p*-Value ^3^		
	CTL	SUP2	SUP4	SUP6	SEM ^2^	TRT	L	Q
4-day Intake, kg d^−1^								
DM								
Hay	3.85 ^§^	3.90	2.80	4.90	1.337	0.2799	0.541	0.167
Liquid supplementation	0	0.61 ^c^	1.16 ^b^	1.52 ^a^	0.088	<0.0001	<0.0001	0.034
Total	3.85	4.51	3.96	6.42	1.407	0.093	0.044	0.229
OM								
Hay	3.75	3.79	2.72	4.76	1.302	0.280	0.546	0.167
Liquid supplementation	0	0.51 ^c^	0.97 ^b^	1.27 ^a^	0.074	<0.0001	<0.0001	0.034
Total	3.75	4.3	3.69	6.03	1.361	0.121	0.068	0.219
CP	0.08 ^d^	0.38 ^c^	0.62 ^b^	0.83 ^a^	0.039	<0.0001	<0.0001	0.160
NDF	3.24 ^b^	3.77 ^ab^	3.30 ^b^	5.26 ^a^	0.755	0.023	0.016	0.126

^1^ CTL= Ad libitum access to LH cv. Floralta. SUP2 = Ad libitum access to LH cv. Floralta and supplementation with 0.9 kg d^−1^ molasses with 32% CP. SUP4 = Ad libitum access to LH cv. Floralta and supplementation with 1.8 kg d^−1^ molasses with 32% CP. SUP6 = Ad libitum access to LH cv. Floralta and supplementation with 2.7 kg d^−1^ molasses with 32% CP. ^2^ Pooled standard error of treatment means, *n* = 6 animals/treatment. ^3^ Treatment effect (TRT); orthogonal contrasts (L and Q) indicate the linear and quadratic effects of the amounts of liquid feed supplementation. ^§^ Averages followed by the same lowercase letter on the same line did not differ from each other in the SAS PDIFF test adjusted by Tukey (*p* > 0.05).

**Table 3 animals-12-02227-t003:** Contribution of dietary N sources to fecal N estimated using δ15N isotopes.

	Treatments ^1^					Contrast *p*-Value ^3^		
	CTL	SUP2	SUP4	SUP6	SEM ^2^	TRT	L	Q
δ^15^N, ‰								
Feces	5.8 ^§^	6.3	6.6	6.3	0.19	0.069	0.064	0.063
Hay	4.3	4.3	4.8	4.6	0.17	0.152	0.128	0.626
Liquid supplementation	0.8	0.8	0.8	0.8	--	--	--	--
% contribution of dietary N in feces								
Hay	100 ^a^	23.3 ^b^	10.4 ^c^	11.5 ^c^	2.41	<0.0001	<0.0001	<0.0001
Liquid supplementation	0 ^c^	76.7 ^b^	89.6 ^a^	88.5 ^a^	2.41	<0.0001	<0.0001	<0.0001

^1^ CTL= Ad libitum access to LH cv. Floralta. SUP2 = Ad libitum access to LH cv. Floralta and supplementation with 0.9 kg d^−1^ molasses with 32% CP. SUP4 = Ad libitum access to LH cv. Floralta and supplementation with 1.8 kg d^−1^ molasses with 32% CP. SUP6 = Ad libitum access to LH cv. Floralta and supplementation with 2.7 kg d^−1^ molasses with 32% CP. ^2^ Pooled standard error of treatment means, *n* = 6 animals/treatment. ^3^ Treatment effect (TRT); orthogonal contrasts (L and Q) indicate the linear and quadratic effects of the amounts of liquid feed supplementation. ^§^ Averages followed by the same lowercase letter on the same line did not differ from each other in the SAS PDIFF test adjusted by Tukey (*p*> 0.05).

**Table 4 animals-12-02227-t004:** Effects of different levels of molasses-based liquid supplementation (32% CP) on the digestibility of diets with limpograss hay ad libitum.

	Treatments ^1^						Contrast *p*-Value ^3^	
	CTL	SUP2	SUP4	SUP6	SEM ^2^	TRT	L	Q
Digestibility, %								
DM	28.5 ^§^	33.4	28.5	25.2	2.08	0.087	0.135	0.064
OM	34.6	38.9	35.7	31.2	1.93	0.085	0.141	0.035
CP	59.2 ^b^	92.1 ^a^	93.8 ^a^	95.0 ^a^	3.59	<0.0001	<0.0001	0.0003
NDF	41.6	45.7	44.7	39.7	1.58	0.056	0.358	0.009
ADF	31.7 ^ab^	36.4 ^a^	35.3 ^a^	26.3 ^b^	1.77	0.004	0.043	0.001

^1^ CTL = Ad libitum access to LH cv. Floralta. SUP2 = Ad libitum access to LH cv. Floralta and supplementation with 0.9 kg d^−1^ molasses with 32% CP. SUP4 = Ad libitum access to LH cv. Floralta and supplementation with 1.8 kg d^−1^ molasses with 32% CP. SUP6 = Ad libitum access to LH cv. Floralta and supplementation with 2.7 kg d^−1^ molasses with 32% CP. ^2^ Pooled standard error of treatment means, *n* = 6 animals/treatment. ^3^ Treatment effect (TRT); orthogonal contrasts (L and Q) indicate the linear and quadratic effects of the amounts of liquid feed supplementation. ^§^ Averages followed by the same lowercase letter on the same line did not differ from each other in the SAS PDIFF test adjusted by Tukey (*p*> 0.05).

**Table 5 animals-12-02227-t005:** Effects of different levels of molasses-based liquid supplementation (32% CP) on the VFA profile with limpograss hay ad libitum.

Item	Treatament ^1^					Contrast *p*-Value ^3^				
	CTL	SUP2	SUP4	SUP6	SEM ^2^	TRT	T	TRT × T	L	Q
VFA, mol/100 mol										
Acetate	71.10	70.77	71.10	70.65	0.262	0.572	0.743	0.145	0.439	0.827
Propionate	16.31	16.66	16.45	16.83	0.160	0.219	0.697	0.025	0.114	0.937
Isobutyrate	0.56	0.46	0.45	0.74	0.074	0.115	0.004	0.021	0.173	0.033
Butyrate	10.21	9.90	9.85	9.61	0.218	0.380	0.165	0.916	0.114	0.878
Isovalerate	0.81 ^b,§^	0.93 ^ab^	0.95 ^a^	0.91 ^ab^	0.028	0.031	0.004	0.005	0.044	0.027
Valerate	0.59	0.73	0.70	0.77	0.041	0.088	0.999	0.212	0.034	0.496
Total VFA, mM	84.70	93.26	91.32	95.65	3.207	0.183	0.274	0.431	0.079	0.531

^1^ CTL = Ad libitum access to LH cv. Floralta. SUP2 = Ad libitum access to LH cv. Floralta and supplementation with 0.9 kg d^−1^ molasses with 32% CP. SUP4 = Ad libitum access to LH cv. Floralta and supplementation with 1.8 kg d^−1^ molasses with 32% CP. SUP6 = Ad libitum access to LH cv. Floralta and supplementation with 2.7 kg d^−1^ molasses with 32% CP. ^2^ Pooled standard error of treatment means, *n* = 6 animals/treatment. ^3^ Treatment effect (TRT); orthogonal contrasts (L and Q) indicate the linear and quadratic effects of the amounts of liquid feed supplementation. ^§^ Averages followed by the same lowercase letter on the same line did not differ from each other in the SAS PDIFF test adjusted by Tukey (*p* > 0.05).

**Table 6 animals-12-02227-t006:** Effects of different levels of molasses-based liquid supplementation (32% CP) on the VFA profile of diets with limpograss hay ad libitum.

Item	Treatment ^1^					Contrast *p*-Value ^3^		
	CTL	SUP2	SUP4	SUP6	SEM ^2^	TRT	L	Q
VFA, mol/100 mol								
Acetate	69.59 ^§^	69.38	68.60	68.33	0.407	0.054	0.011	0.928
Propionate	20.52 ^b^	20.82 ^b^	21.59 ^ab^	22.10 ^a^	0.353	0.013	0.002	0.684
Isobutyrate	0.65 ^a^	0.63 ^a^	0.62 ^a^	0.58 ^b^	0.013	0.001	0.0002	0.107
Butyrate	7.35	7.30	7.40	7.28	0.053	0.104	0.369	0.320
Isovalerate	1.11 ^a^	1.11 ^a^	1.06 ^a^	0.95 ^b^	0.002	0.002	0.001	0.016
Valerate	0.76	0.75	0.73	0.76	0.022	0.765	0.766	0.427
Total VFA, mM	88.42 ^b^	90.71 ^b^	101.88 ^a^	105.24 ^a^	2.213	0.003	0.001	0.801

^1^ CTL = 100% LH, SUP2 = 85% LH and 15% liquid supplementation, SUP4 = 70% LH and 30% liquid supplementation, SUP6 = 55% LH and 45% liquid supplementation. ^2^ Grouped standard error of treatment means, *n* = 4 bottles/treatment. ^3^ Treatment effect (TRT); orthogonal contrasts (L and Q) indicate the linear and quadratic effects of the amounts of liquid feed supplementation. ^§^ Averages followed by the same lowercase letter on the same line did not differ from each other in the SAS PDIFF test adjusted by Tukey (*p* > 0.05).

## Data Availability

The data presented in this study are available on request from the corresponding author. The data are not publicly available due to internal policy.

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
