# Peer review of "Supplementation of Molasses-Based Liquid Feed for Cattle Fed on Limpograss Hay"

_animals, 2022, doi:10.3390/ani12172227_

Round 1

Reviewer 1 Report

COMMENTS TO THE AUTHORS:

This manuscript describes the results of a randomized complete block design evaluating the supplementation of increasing levels (4 treatments) of a liquid feed containing molasses and urea to a basal diet containing CP deficient Hemarthria altissima. The manuscript has some deficiencies in terminology, English standards, study description and design that I will outline below.

MAJOR CONCERNS:

There are 3 major concerns:

1.      The study only had 6 animals per treatment, which probably limited the ability to detect significant differences among treatments (increased risk of error type II). This needs to be addressed in the discussion and should limit the scope of your conclusions accordingly.

2.      The manuscript does not provide any information about the liquid molasses + urea. This is the core of the study: if the mix was prepared on site, how was this supplement prepared? If commercial product, it needs to properly identify the manufacturer. Also, the fact that the analysis shows 45.5% CP but the authors indicate molasses was supplemented with 32% CP needs to be explained.

3.      This manuscript is missing a major piece of the puzzle: why is there no information about the effect of treatments on animal weights?

MINOR CONCERNS:

The following are aspects that need to be addressed:

  1. The manuscript reads as if it is a thesis dissertation chapter and could use some cleanup/simplification to reduce the number of pages.
  2. Throughout the manuscript:
    1. Authors need to standardize the use of numbers or citations.
    2. Add space between sentences/words and citation number.
    3. Add space between number and units.
    4. Limit the number of digits to 3 decimals or 3 significant figures throughout the manuscript.
    5. Add 0 before the decimal points in your tables: <.001 should be <0.001.
  3. Page 1, line 14: I suggest changing from “pleasant taste” to “palatability”.
  4. Page 2, line 66: replace “, although it offers an opportunity to stimulate ingestion,” with “as”.
  5. Page 3, line 106-107: clarify if animals were housed individually or in groups.
  6. Page 7, Table 2: Table 2 suggests %CP in the DM ingested was 2.1, 8.4, 15.7 and 12.9. Please, address that in your discussion.
  7. Page 7, Table 3: what are the units for δN15? This data appear to suggest that the urea-N added in the molasses supplement was poorly utilized. BUN could have supported that and in vitro N-NH3 suggests that is the case.
  8. Page 8, Table 4: Why is there no column for TRT in this table? What do the asterisks on ADF row represent?
  9. Page 10, lines 375-379: this sentence is too long. Please, consider splitting it in multiple sentences
  10. Page 14, “Ruminal pH”: This results section is included under Experiment 2, suggesting it is in vitro rumen fluid pH. If that is the case, please, change the section title and subsequent contents to reflect “in vitro rumen fluid pH”.
  11. Page 14, line 448: replace the sentence “the elevation of offered levels of molasses-based liquid supplement (32% CP)” with “molasses-based liquid supplement levels”.
  12. Page 17, line 483: replace the word “elevation” with “increase”.
  13. Page 17, lines 502-505: this sentence suggests supplementation improved “the use of the forage fibrous fraction by the ruminal microbiota”, but NDF and ADF results do not support that assertion.
  14. Page 18, lines 527-528: “Digestibility without the presence of molasses…” Digestibility of what fraction?
  15. Page 18, lines 532-533: “For ADF, the control treatments and those with the highest supplementation level had a decrease in relation to the others”. Had a decrease in what?
  16. Page 19, lines 597-600: Another long, confusing sentence. Please, consider splitting in multiple sentences.
  17. Page 19, line 613: “In present study” should be “In the present study”.
  18. Page 20, line 631: What does “attenuating drops” mean? Please, consider using different terminology.
  19. Page 20, line 661: replace “range” with “ranging”.
  20. Page 20, Conclusions: consider limiting the scope of your conclusions. You had limited number of animals and that may have played an important role in your statistical findings.

Author Response

Reviewer 1

COMMENTS TO THE AUTHORS:

This manuscript describes the results of a randomized complete block design evaluating the supplementation of increasing levels (4 treatments) of a liquid feed containing molasses and urea to a basal diet containing CP deficient Hemarthria altissima. The manuscript has some deficiencies in terminology, English standards, study description and design that I will outline below.

Thanks for your comments. We addressed specifically each item raised during the review.

MAJOR CONCERNS:

There are 3 major concerns:

  1. The study only had 6 animals per treatment, which probably limited the ability to detect significant differences among treatments (increased risk of error type II). This needs to be addressed in the discussion and should limit the scope of your conclusions accordingly.

We understand the issue raised by the reviewer, but we respectfully disagree. We run the power analysis for digestibility and intake variables, and they were above 0.99 (decreasing risk of type II error). More animals would be better, for sure, but we do not think our conclusions were affected by our number of animals.

  1. The manuscript does not provide any information about the liquid molasses + urea. This is the core of the study: if the mix was prepared on site, how was this supplement prepared? If commercial product, it needs to properly identify the manufacturer. Also, the fact that the analysis shows 45.5% CP but the authors indicate molasses was supplemented with 32% CP needs to be explained.

We added information for the liquid feed. This is a commercial product. As for the %CP, the liquid feed normally presents values as fed, that is, around 32%. Normally, for liquid feed we use as fed basis and not DM (45% on a DM basis). However, to be clear, we have added the information in the manuscript. We also added information on the product and manufacturer (lines 112-115).

  1. This manuscript is missing a major piece of the puzzle: why is there no information about the effect of treatments on animal weights?

The objective of this trial was not to study weight gain. The design with a shorter period would not allow us to accurately measure weight gains. Our objective was to measure intake and digestibility of nutrients. Animals were weighed only twice in the entire experiment, one day before the start of the trial and at the end of the trial. The weighing events performed were not sufficient to assess weight gain. Therefore, there is no weight gain data in the manuscript.

MINOR CONCERNS:

The following are aspects that need to be addressed:

  1. The manuscript reads as if it is a thesis dissertation chapter and could use some cleanup/simplification to reduce the number of pages.

We performed a review and try to address your comments. We don’t think the manuscript is too long as you are suggesting.

  1. Throughout the manuscript:
    1. Authors need to standardize the use of numbers or citations.

Revised. Now it is standardized.

    1. Add space between sentences/words and citation number.

The correction has been made.

    1. Add space between number and units.

The correction has been made.

    1. Limit the number of digits to 3 decimals or 3 significant figures throughout the manuscript.

The correction has been made.

    1. Add 0 before the decimal points in your tables: <.001 should be <0.001.

The correction has been made.

  1. Page 1, line 14: I suggest changing from “pleasant taste” to “palatability”.

The correction has been made.

  1. Page 2, line 66: replace “, although it offers an opportunity to stimulate ingestion,” with “as”.

The correction has been made.

  1. Page 3, line 106-107: clarify if animals were housed individually or in groups.

This description is in lines 116-117

  1. Page 7, Table 2: Table 2 suggests %CP in the DM ingested was 2.1, 8.4, 15.7 and 12.9. Please, address that in your discussion.

This subject is covered in lines 490-493.

  1. Page 7, Table 3: what are the units for δN15? This data appear to suggest that the urea-N added in the molasses supplement was poorly utilized. BUN could have supported that and in vitro N-NH3 suggests that is the case.

We inserted the unit of δ15N in the table (‰). True, BUN could have supported this.

  1. Page 8, Table 4: Why is there no column for TRT in this table? What do the asterisks on ADF row represent?

Revised. Thanks for catching that. That information was missing.

  1. Page 10, lines 375-379: this sentence is too long. Please, consider splitting it in multiple sentences.

The correction has been made.

  1. Page 14, “Ruminal pH”: This results section is included under Experiment 2, suggesting it is in vitro rumen fluid pH. If that is the case, please, change the section title and subsequent contents to reflect “in vitro rumen fluid pH”

The correction has been made.

  1. Page 14, line 448: replace the sentence “the elevation of offered levels of molasses-based liquid supplement (32% CP)” with “molasses-based liquid supplement levels”.

The correction has been made.

  1. Page 17, line 483: replace the word “elevation” with “increase”.

The correction has been made.

  1. Page 17, lines 502-505: this sentence suggests supplementation improved “the use of the forage fibrous fraction by the ruminal microbiota”, but NDF and ADF results do not support that assertion.

Apologies for our mistake. We had not put the P values for the treatment in table 4. I made the correction and there is a quadratic effect for ADF, indicating that the increase in supplementation modified the digestibility for ADF.

  1. Page 18, lines 527-528: “Digestibility without the presence of molasses…” Digestibility of what fraction?

Our mistake. The information is missing. In this sentence, it concerns the digestibility of CP. I made the correction in the text.

  1. Page 18, lines 532-533: “For ADF, the control treatments and those with the highest supplementation level had a decrease in relation to the others”. Had a decrease in what?

The topic of this discussion is digestibility of diets. In this case, it is the digestibility of ADF. I corrected the sentence to make it clearer.

  1. Page 19, lines 597-600: Another long, confusing sentence. Please, consider splitting in multiple sentences.

The correction has been made.

  1. Page 19, line 613: “In present study” should be “In the present study”.

The correction has been made.

  1. Page 20, line 631: What does “attenuating drops” mean? Please, consider using different terminology.

The correction has been made.

  1. Page 20, line 661: replace “range” with “ranging”.

The correction has been made.

  1. Page 20, Conclusions: consider limiting the scope of your conclusions. You had limited number of animals and that may have played an important role in your statistical findings.

As we indicated before, we run the power analysis and our results are correct.

Reviewer 2 Report

The comments are added in the file please

Author Response

Reviewer 2 put his comments in the PDF file. 

We addressed all the comments from the reviewer, and it is incorporated in the new revised version.

Comments were mainly related to style and formatting.

Round 2

Reviewer 1 Report

A major concern remains:

The study only had 6 animals per treatment. Respectfully, I understand you ran a power analysis that supported your study, but unfortunately your results and p-values paint a different picture. Digestibility, for instance: except for CP digestibility, all other digestibility variables were not significantly different, but trends that would possibly be significant had the N in this study been greater. ADF digestibility had a TRT p-value of 0.004, but the control did not differ from the other treatments (as indicated by the letters a,b next to the control value), in spite of the discussion on ADF digestibility suggests. Instead, ADF digestibility was lower with the highest level of supplementation compared with the other 2 lower levels of supplementation (not compared with control), a possible result of 25% more hay intake in that latter treatment. In spite of the large numerical difference, hay DM intake did not differ across all treatments. The authors found significant increase in SUP6 NDF intake: if ADF intake follows a similar pattern to NDF intake, that could help explain the ADF digestibility results.

Furthermore, the authors concluded that digestibility is not significantly different among treatments, in contrast to statements in the discussion.

It is also confusing to suggest that the liquid feed improved nutrient supply: the in vivo study results do not strongly support this assertion.

Please, adjust the conclusions and be specific about the source of your assertions.

MINOR CONCERNS:

The following are aspects that need to be addressed:

  1. Some of the rounded numbers are incorrectly rounded or had typos. Please, carefully review them.

2.      The term “the elevate” is often used to mean the increase or the rise. It is not a commonly used terminology. Consider changing.

3.      Some tables and figures have asterisks. They need to be identified/explained in the figure/table description for clarity.

4.      The term “liquid Feed supplementation” is used in the footnotes throughout. Use “liquid feed supplementation”.

Author Response

Authors' responses to reviewers

Dear Editor,

Thanks for considering our manuscript for publication. We addressed all issues raised during the second round of the review process and they are specifically listed below. Should you have further questions, please do not hesitate in contacting us.

Regards

Jose Dubeux (corresponding author)     

Round 2

Reviewer 1

A major concern remains:

The study only had 6 animals per treatment. Respectfully, I understand you ran a power analysis that supported your study, but unfortunately your results and p-values paint a different picture. Digestibility, for instance: except for CP digestibility, all other digestibility variables were not significantly different, but trends that would possibly be significant had the N in this study been greater. ADF digestibility had a TRT p-value of 0.004, but the control did not differ from the other treatments (as indicated by the letters a,b next to the control value), in spite of the discussion on ADF digestibility suggests. Instead, ADF digestibility was lower with the highest level of supplementation compared with the other 2 lower levels of supplementation (not compared with control), a possible result of 25% more hay intake in that latter treatment. In spite of the large numerical difference, hay DM intake did not differ across all treatments. The authors found significant increase in SUP6 NDF intake: if ADF intake follows a similar pattern to NDF intake, that could help explain the ADF digestibility results.

Furthermore, the authors concluded that digestibility is not significantly different among treatments, in contrast to statements in the discussion.

It is also confusing to suggest that the liquid feed improved nutrient supply: the in vivo study results do not strongly support this assertion.

Please, adjust the conclusions and be specific about the source of your assertions.

We understand the issue raised by the reviewer. The conclusion was modified following the reviewer's suggestion.

MINOR CONCERNS:

The following are aspects that need to be addressed:

  1. Some of the rounded numbers are incorrectly rounded or had typos. Please, carefully review them.

Apologies for our mistake. The correction has been made.

  1. The term “the elevate” is often used to mean the increase or the rise. It is not a commonly used terminology. Consider changing.

The change has been made

  1. Some tables and figures have asterisks. They need to be identified/explained in the figure/table description for clarity.

The correction has been made.

  1. The term “liquid Feed supplementation” is used in the footnotes throughout. Use “liquid feed supplementation”.

Thanks for catching that. The correction has been made.
